# Photo-mediated selective deconstructive geminal dihalogenation of trisubstituted alkenes

Han Wang[1,4], Ren Wei Toh[1,4], Xiangcheng Shi[1], Tonglin Wang[2], Xu Cong[1] & Jie Wu [1,3✉]

Selective deconstructive functionalization of alkenes, other than the well-established olefin metathesis and ozonolysis, to produce densely functionalized molecular scaffolds is highly attractive but challenging. Here we report an efficient photo-mediated deconstructive germinal dihalogenation of carbon-carbon double bonds. A wide range of geminal diiodoalkanes and bromo(iodo)alkanes (>40 examples) are directly prepared from various trisubstituted alkenes, including both cyclic and acyclic olefins. This C=C cleavage is highly chemoselective and produces geminal dihalide ketones in good yields. Mechanistic investigations suggest a formation of alkyl hypoiodites from benzyl alcohols and $N$-iodoimides, which undergo light-induced homolytic cleavage to generate active oxygen radical species.

[1] Department of Chemistry, National University of Singapore, 3 Science Drive 3, Singapore 117543, Republic of Singapore. [2] College of Chemistry and Chemical Engineering, Northwest Normal University, 730070 Lanzhou, Gansu, China. [3] National University of Singapore (Suzhou) Research Institute, 377 Lin Quan Street, Suzhou Industrial Park, 215123 Suzhou, Jiangsu, China. [4] These authors contributed equally: Han Wang, Ren Wei Toh. ✉email: chmjie@nus.edu.sg

n organic synthesis, common functionalization usually focuses on the installation or modification of functional groups without significantly changing the backbones of molecules. In stark contrast, deconstructive functionalization is attractive as it can drastically change the scaffolds of molecules to introduce new chemical space, unmask dormant functional groups, and create functionalities tethered at a predefined distance determined by ring sizes of the reactants.

The carbon-carbon double bond is one of the most fundamental functionalities in organic molecules. Various methods have been developed to convert alkenes to important intermediates and fine chemical products, which play vital roles in the fields of material science, biochemistry, pharmaceutical science, and the chemical industry[1–8]. Deconstructive functionalization of alkenes has been well developed to introduce two functional groups at different sites of olefins (Fig. 1a). For instance, transition-metal-catalyzed C=C bond cleavage processes, such as olefin metathesis, have found wide application in natural product and material synthesis[9–12]. Ozonolysis and other similar oxidations with various organic and inorganic oxidants were robust to introduce two carbonyl derivatives from a single C=C bond[13–17]. Aside from such well-established strategies, other types of deconstructive functionalization of alkenes producing densely functionalized scaffolds remain rare and challenging[18–27].

Organohalides are versatile building blocks in synthetic chemistry. They are widely utilized as precursors in transition-metal-catalyzed cross-coupling, radical reactions, nucleophilic substitutions, and metal-halide exchanges[28–30]. Among them, geminal dihalides represent a unique class of compounds and have been used as carbene precursors and multi-functional synthons. However, efficient synthetic pathways to synthesize geminal dihalides are quite limited, which significantly restricts the investigation and wide application of this unique family of compounds[31–35]. Herein, we report a direct synthetic route to geminal dihalides by photo-mediated deconstructive fragmentation of cyclic or acyclic trisubstituted alkenes (Fig. 1b).

## Results

**Reaction optimization.** We initially designed a cascade hydro-halogenation of trisubstituted alkenes and subsequent photo-induced β-scission of the generated alcohol intermediates. Based on previous reports from groups of Chen and Zhu on photo-mediated conversion of alcohols to oxygen radicals[36–42], our study commenced by using 1-phenyl-1-cyclohexene (**1**) as a model substrate. As illustrated in Table 1, treatment of **1** with eosin Y (1 mol%) as the photocatalyst, *N*-iodosuccinimide (NIS, 4 equiv), H$_2$O (50 equiv) and acetoxyl benziodoxole (BIOAc, 2 equiv) in MeCN under blue light-emitting diode (LED) irradiation, the desired product, 6,6-diiodo-1-phenylhexan-1-one (**2**) was obtained in 40% yield (entry 1). Further investigation revealed that a similar result could be obtained in the absence of any photocatalyst (entry 2). To our surprise, **2** could be generated in 28% yield even without BIOAc (entry 3). A moderate temperature (50 °C) significantly accelerated the reaction (entries 4–5). Evaluation of solvents indicated that a mixed solvent of EtOAc/MeNO$_2$ (10:1) was the optimal choice, leading to the generation of **2** in 81% yield (entries 6–9). Using 1,3-diiodo-5,5-dimethyl-hydantoin (DIH) as the iodination agent afforded an improved yield compared to that with NIS (entry 10). Light irradiation was essential as no product **2** could be detected in the absence of light (entry 11).

**Substrate scope.** With the optimized conditions established, we investigated the scope of the deconstructive geminal diiodination of cyclic alkenes. As shown in Fig. 2, a variety of aryl-substituted cyclohexene derivatives underwent deconstructive geminal diiodination effectively to deliver products **2–15** in moderate to good yields. Cyclohexenes containing aryl rings with electronically distinct substituents in the *ortho-*, *meta-* or *para*-position afforded products **2–10** in similar yields, with the exception of a

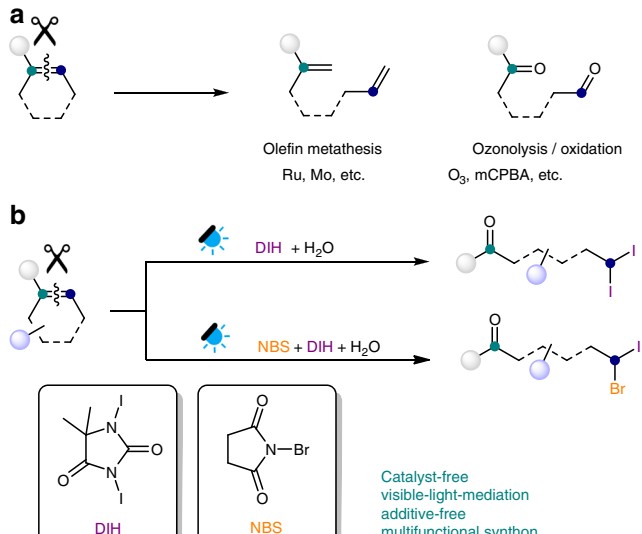

**Fig. 1 Deconstructive functionalization of alkenes. a** Common strategies for C=C bond cleavage. **b** Visible-light-mediated deconstructive oxidative geminal dihalogenation of trisubstituted alkenes (this work). DIH 1,3-diiodo-5,5-dimethyl-hydantoin, NBS *N*-bromosuccinimide, mCPBA *meta*-chloroperoxybenzoic acid.

### Table 1 Optimization of oxidative deconstructive geminal diiodination[a].

PC (1 mol%), NIS (4 equiv), H$_2$O (50 equiv),
Additive (x equiv), solvent (0.1 M),
argon, blue LED (80 W), 36 h

| entry | catalyst | solvent | additive (x equiv) | yield (%)[b] |
|---|---|---|---|---|
| 1 | eosin Y | MeCN | BIOAc (2) | 40 |
| 2 | – | MeCN | BIOAc (2) | 41 |
| 3 | – | MeCN | – | 28 |
| 4[c] | – | MeCN | – | 60 |
| 5[d] | – | MeCN | – | 41 |
| 6[c] | – | DCE | – | 34 |
| 7[c] | – | EtOAc | – | 79 |
| 8[c] | – | acetone | – | n.d. |
| 9[c] | – | EtOAc/ MeNO$_2$ (10:1) | – | 81 |
| 10[c,e] | – | EtOAc/ MeNO$_2$ (10:1) | – | 91(84)[f] |
| 11[c,e,g] | – | EtOAc/ MeNO$_2$ (10:1) | – | n.d. |

NIS N-iodosuccinimide, DCE 1,2-dichloroethane, n.d. not determined.
[a]Standard conditions: **1** (0.2 mmol), NIS (0.8 mmol), and H$_2$O (10 mmol) in solvent (0.1 M), irradiated under blue LED lamps (80 W) for 36 h at 30 °C.
[b]Yields determined by analysis of the crude $^1$H-NMR spectra using dibromomethane as an internal standard.
[c]Reaction was performed at 50 °C.
[d]Reaction was performed at 80 °C.
[e]DIH (0.4 mmol, 2 equiv) was used instead of NIS.
[f]Isolated yields.
[g]No light irradiation.

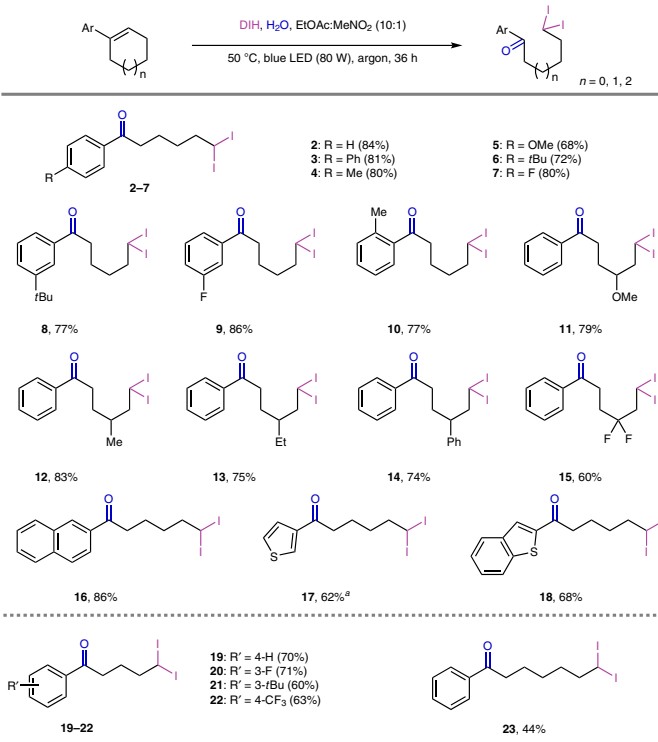

**Fig. 2 Scope of oxidative deconstructive geminal diiodination of cyclic alkenes.** Isolated yields unless otherwise indicated. Performed with alkene (0.2 mmol), $H_2O$ (10 mmol), DIH (0.4 mmol) in EtOAc:MeNO$_2$ (10:1), irradiated under blue LED lamps (80 W) for 36 h at 50 °C. $^a$4 equiv NIS instead of DIH.

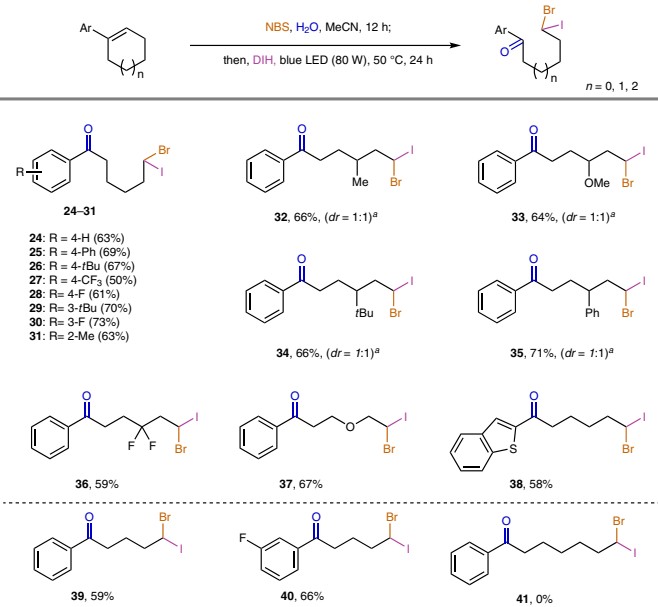

**Fig. 3 Scope of oxidative deconstructive geminal bromo-iodination of cyclic alkenes.** Isolated yields unless otherwise indicated. Performed with alkene (0.2 mmol), NBS (0.21 mmol), $H_2O$ (10 mmol), DIH (0.3 mmol) in MeCN. See Supplementary Information for experimental details. $^a$Dr values were determined by analysis of $^1$H NMR spectra of the crude product mixture.

compound bearing a strong electron-donating methoxy substituent, with which electrophilic aryl iodination occurred in the presence of DIH, resulting in a lower yield of product **5**. Various substituents on the cyclohexene ring, including methoxy, alkyl, phenyl, difluoro, were well tolerated, giving products **11–15** with similar yields. Naphthyl and hetero-aryl-substituted cyclohexenes were also suitable substrates, and underwent ring-opening and geminal diiodination to deliver compounds **16–18** with yields of 62–86%. Because the thiophene ring is very reactive towards electrophilic substitution, NIS was applied in place of DIH to avoid the iodination of the thiophene ring in the synthesis of **17**. The deconstructive geminal diiodination is not limited to cyclohexene derivatives; a range of aryl-substituted cyclopentene and cycloheptene derivatives were all viable in this deconstructive functionalization to achieve products **19–23** in moderate to good yields. Variation of the size of the ring containing the double bond led to geminal diiodide ketone products bearing carbon chains with varying lengths.

Besides the use of DIH to generate geminal diiodo products, this strategy can be further expanded to produce potentially more useful bromo(iodo)alkanes by modifying the optimal reaction conditions by addition of NBS (Fig. 3). This cascade transformation was applied to a variety of aryl-substituted cyclic alkenes to afford the corresponding geminal bromo-iodination products. A variety of functional groups on the aryl rings or the cyclohexene scaffolds were well tolerated and delivered products **24–36** in moderate to good yields. 3,6-Dihydropyran was a good substrate for this cascade reaction thus further enriching the skeletal diversification (**37**). The robustness of this strategy was further explored with heteroaryl rings and cyclopentenes which generated the corresponding bromo-iodide arylketone products **38–40** in moderate yields. However, 1-phenyl-1-cycloheptene did not

afford the corresponding bromo(iodo)alkane product **41** under our reaction conditions.

As illustrated in Fig. 4a, trialkyl-substituted alkene **42** was subjected to the standard conditions for geminal diiodination and bromo-iodination, but no corresponding product could be generated. A high temperature (80 °C) could not promote geminal diiodination of **42** either, while bromo-iodination product **43** could be isolated in 21% yield. However, further increasing the temperature to 100 °C afforded only a trace amount of **43**. Control experiments (Supplementary Fig. 2) indicated the instability of halohydrins at high temperatures and the ineffectiveness of trialkyl-substituted halohydrins in the deconstructive iodination step. These results highlighted the importance of a suitable temperature and aromatic substituents for the success of the deconstructive geminal dihalogenation. To further expand the scope of this reaction, acyclic trisubstituted olefins were evaluated. As shown in Fig. 4b, acyclic alkenes **44–46** with different substituent patterns were subjected to the optimal diiodination conditions, and only the gem-alkyl-aryl-substituted olefin **46** delivered the corresponding diiodide product **47** in a useful yield (47%). Further tuning of the electronic properties in the vinyl aryl substituent (**48** and **49**) resulted in product **47** with similar yields. The deconstructive geminal diiodination proceeded readily with alkenes bearing different remote functionalities (**50** and **52**), affording products **51** and **53**, respectively, in moderate yields.

**Mechanistic elucidation with supporting evidence.** Various control experiments were performed to elucidate the reaction mechanism. First, **1** was treated with DIH or NBS and water in the absence of light to afford iodohydrin **54** or bromohydrin **55**, respectively. Tertiary alcohols **54** and **55** were subjected to the standard light-promoted conditions and afforded products **2** and **24**, respectively, but gave no product in the absence of light (Fig. 5a). These control experiments showed that the iodohydrin and bromohydrin were intermediates in the cascade one-pot

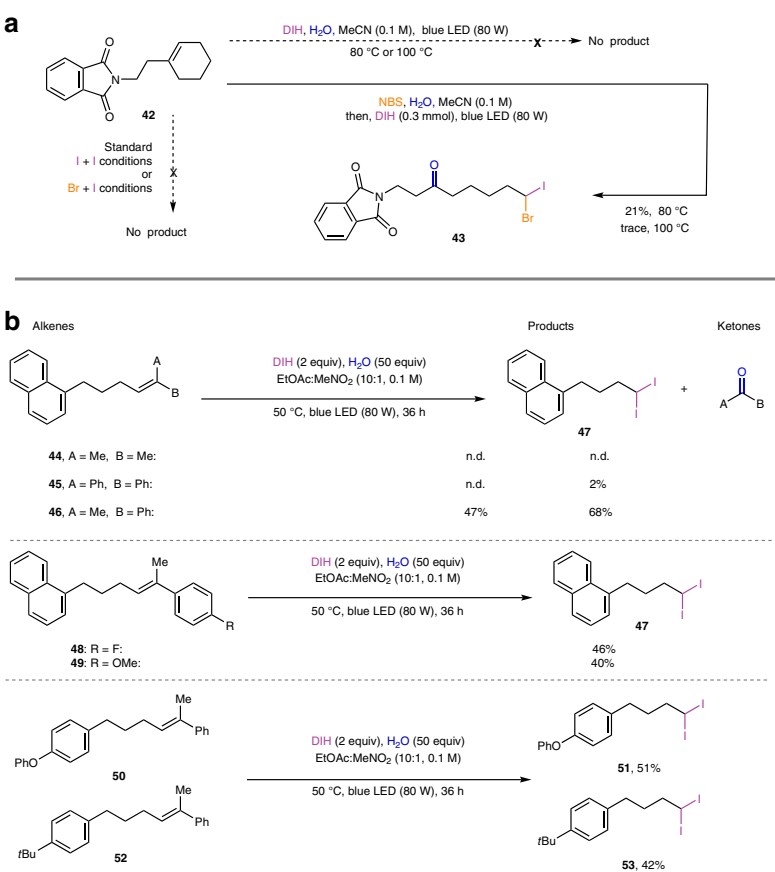

**Fig. 4 Investigation of other trisubstituted alkenes. a** Trialkyl-substituted cyclohexenes. **b** Acyclic trisubstituted alkenes.

reaction. When TEMPO was added into the reaction mixture, no dihalide product could be detected, which supported a radical based reaction mechanism. ESR measurements also indicated the presence of radical species when reacting DIH with **55** in MeCN under light irradiation (Supplementary Figs. 9 and 10).

Based on previous literatures[43], alcohols can react with acetyl hypoiodite (AcOI) to generate alkyl hypoiodites. The O-I bonds of the alkyl hypoiodites could undergo homolytic cleavage under light irradiation to deliver a transient alkoxy radical species. We, therefore, prepared **56** by treating **55** with acetyl hypoiodite in dark, and generation of the unstable intermediate **56** was confirmed by $^1$H NMR analysis of the reaction mixture (Supplementary Fig. 3). Irradiation of **56** formed in situ with blue LED lamps for 10 min afforded product **24** in 26% yield. When a mixture of **55** and DIH in MeCN, reacting in the dark, was examined by $^1$H NMR and GC-MS analysis, **56** and 1-iodo-5,5-dimethylhydantoin were detected (Supplementary Fig. 4–6). This result indicates the generation of an alkyl hypoiodite from a phenyl-substituted alcohol and DIH. Even though we cannot fully exclude the possibility of formation of an electron donor-acceptor (EDA) complex between **55** and DIH, UV-Vis measurements of different stoichiometry between **55** and DIH did not follow the curve of Job's plot, which is normally observed for an EDA complex (Fig. 5b). Light on/off experiments indicated that a long radical chain process is unlikely. As illustrated in Fig. 5c, the product formation ceased immediately when the light source was periodically switched off and resumed when light was turned on.

Based on existing related literatures[43–47] and all the experimental results described above, a plausible mechanism was proposed and is described in Fig. 5d. The cascade reaction is initiated by hydroxyhalogenation of alkene **I** in the presence of water and an electrophilic halogen source[44]. The resulting halohydrin **II** reacts with DIH to deliver the alkyl hypoiodite intermediate **III**. Light irradiation induces the homolysis of the labile I–O bond to generate the reactive oxygen radical species **IV**[43,45,46]. Carbon radical **V** is formed by β-scission of radical **IV** and subsequent iodination with the iodine radical or DIH accomplishes product **VI**[43,46–51].

To further support the proposed ring-opening iodination, density functional theory (DFT) calculations were conducted on the model reaction of **55** with DIH. The resulting energy profiles of the reaction process are displayed in Fig. 6. The complexation of **55** and DIH leads to a zwitterionic intermediate **Int1** through **TS1** with active free energy of 32.6 kcal/mol. This relatively high active free energy may explain why an elevated temperature is essential for an efficient transformation (Table 1, entry 3 vs 4). The iodine atom between the two carbonyl groups is more electrophilic than the other iodine atom in DIH, and undergoes complexation with alcohol **55**. The subsequent heterolysis of the O-I bond occurs together with the alcohol deprotonation through a transition state **TS2** to give **56** and dimethyliodohydantoin with a barrier of 24.4 kcal/mol relative to **Int1**. Under blue light excitation, the ground-state **56** (**S$_0$**) is pumped to the first singlet excited-state **56*** (**S$_1$**) with excitation energy of 66.3 kcal/mol (2.87 eV) according to time-dependent density functional theory (TD-DFT) calculations. Homolytic cleavage of the O-I bond in **56*** produces the active oxygen radical **Int2**. The subsequent β-scission is a facile process with a barrier of 8.5 kcal/mol to give the transient carbon radical **Int3**, followed by exergonic trapping with the iodine radical to accomplish the generation of product

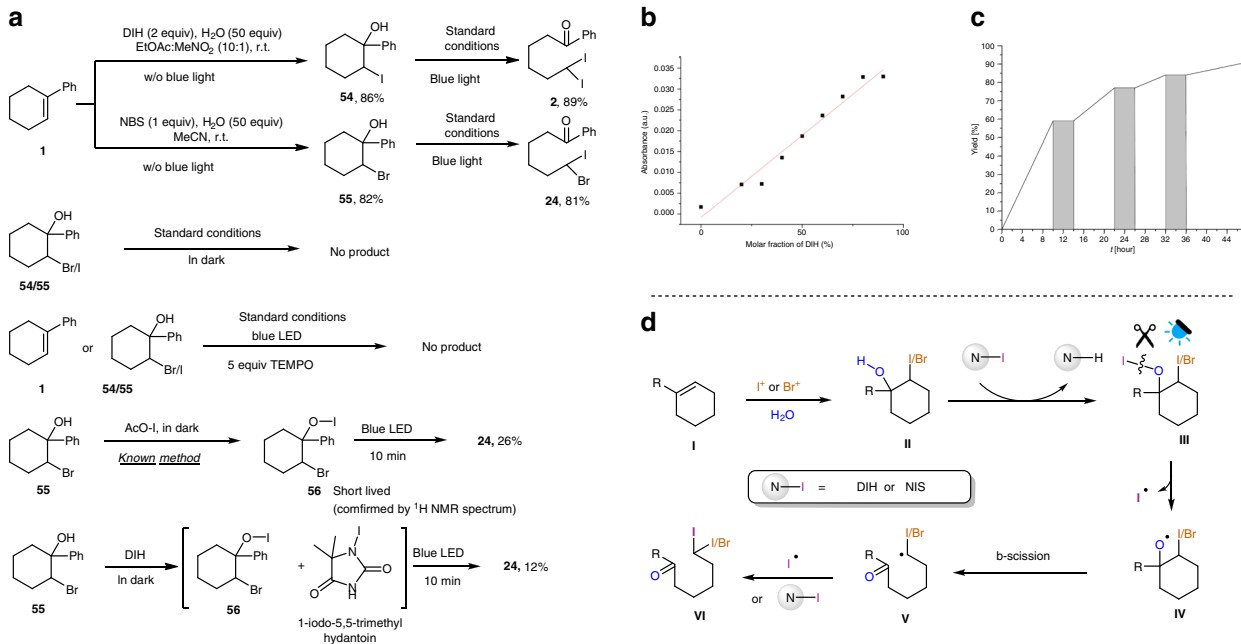

**Fig. 5 Plausible mechanism with supporting evidence. a** Control experiments. **b** UV-Vis measurements of stoichiometry between **55** and DIH @ 450 nm. **c** Light on-off experiments of deconstructive geminal diiodination of **1**. **d** Plausible mechanism.

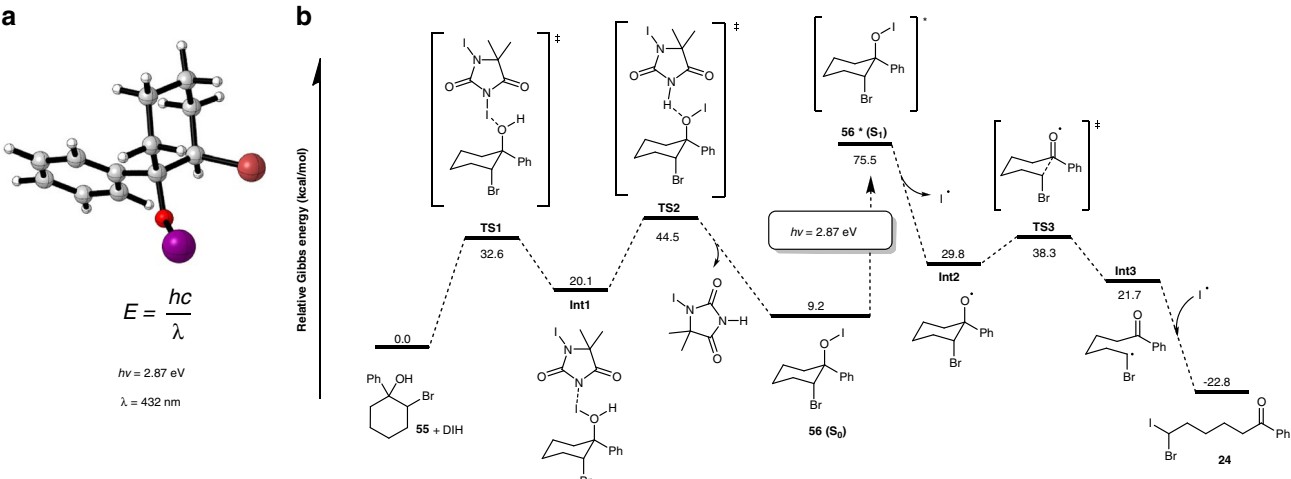

**Fig. 6 DFT calculations for the ring-opening iodination of 55 with DIH. a** Optimized geometry of **55** based on DFT calculation. **b** Free energy profiles for the ring-opening iodination.

**24**. The geometry of the unstable alkyl hypoiodite **56** was optimized and the calculated maximum light absorption is at 432 nm (Fig. 6a), which is close to the blue LED maximum emission (456 nm) used in this study.

**Further synthetic applications**. The oxidative geminal diiodination reaction could be achieved on gram quantities by prolonging the reaction time (Fig. 7a). To further demonstrate the synthetic utility of our methods, derivatization of generated geminal dihalide products was attempted (Fig. 7b). The diiodo compound **2** could be easily converted to disubstituted alkene **57** with moderate stereoselectivity by Takai-Utimoto olefination[52,53]. Base-promoted elimination of **2** led to vinyl iodide **58** in excellent yield with a moderate *E/Z* selectivity[54]. Synthetically useful geminal bis(boronate) (**59**) was easily produced by subjecting the diiodo compound **2** to copper-catalyzed boronation[55]. A photoredox catalyzed deiodination using Hantzsch ester as the

hydrogen source reduced **2** to **60** bearing an unsubstituted alkyl chain. More interestingly, highly selective derivatization of the iodide in the bromo(iodo)alkane products could be achieved. The photoredox reaction using Hantzsch ester converted **24** to alkylbromide **61** in 90% yield. Based-promoted elimination of **24** selectively produced the corresponding vinyl bromide **62**. Bromoiodide **24** could undergo a selective $S_N2$ reaction with sodium azide to afford **63** in 86% yield. Subjection of this azide (**63**) to copper-catalyzed azide-alkyne cycloaddition afforded a bromo triazole[56], which could undergo further nucleophilic substitution to deliver *α*-functionalized triazole **64** in 69% yield.

## Discussion

In summary, we herein report a simple protocol for deconstructive geminal diiodination and bromo-iodination of trisubstituted alkenes under visible light irradiation. The success of this transformation relies on the formation of a labile alkyl hypoiodite

**Fig. 7 Gram-scale synthesis and further synthetic diversification. a** Gram-scale synthesis. **b** Synthetic diversification. B$_2$Pin$_2$ bis (pinacolato)diboron, DBU 1,8-diazabicyclo (5.4.0)undec- 7-ene.

intermediate between the halohydrin and DIH. The protocol is distinguished by its operational simplicity, metal-free and catalyst-free characters, a wide scope of both cyclic and acyclic alkenes, delivery of useful and otherwise difficultly accessible synthons, and controllable chain length of products by choosing alkenes with different ring sizes.

## Methods

**General procedure of the deconstructive geminal diiodination**. Alkene (0.2 mmol, 1 equiv) and 1,3-diiodo-5,5-dimethylhydantoin (0.4 mmol, 2 equiv), H$_2$O (10 mmol, 50 equiv), and EtOAc:MeNO$_2$ (10:1, 2 mL) were added to a schlenk tube (10 mL) equipped with a magnetic stirring bar. Then, the reaction mixture was operated by freeze-pump-thaw procedures three times and backfilled with argon. The resulting solution was irradiated by blue LED lamps (2 × 40 W) and magnetically stirred at 50 °C. After 36 h, the reaction solution was concentrated, and the product was purified by column chromatography (SiO$_2$). The diastereomeric ratio was determined by $^1$H NMR of the crude product mixture. See Supplementary Methods for details.

**General procedure of the deconstructive bromo-iodination**. Alkene (0.2 mmol) and N-bromosuccinimide (0.21 mmol, 1 equiv), H$_2$O (10 mmol, 50 equiv), and MeCN (2 mL) were added to a schlenk tube (10 mL) equipped with a magnetic stirring bar. The reaction mixture was operated by freeze-pump-thaw procedures for three times and backfilled with argon. The resulting solution was magnetically stirred at 50 °C for 12 h. Then, 1,3-diiodo-5,5-dimethylhydantoin (0.3 mmol, 1.5 equiv) was added to the reaction mixture under argon. The reaction mixture was irradiated by blue LED lamps (2 × 40 W) and magnetically stirred at 50 ºC. After 24 h, the reaction solution was concentrated, and the product was purified by column chromatography (SiO$_2$). The diastereomeric ratio was determined by $^1$H NMR of the crude product mixture. See Supplementary Methods for details.

**Computational details**. Density functional theory (DFT) calculations were performed for the verification of the mechanism. The geometries optimization in this study (except **56** and **56***) was performed at the (u)B3LYP-D3(BJ) level of theory. The 6–311 + g(d,p) basis set was used for all H, C, N, and O atoms, and the Stuttgart–Dresden basis set (SDD) was employed for Br and I atoms. The nature of the stationary points (minima with no imaginary frequency or transition states with one imaginary frequency) was confirmed. The free energies of the optimized geometries were calculated at the same level of theory, taking into account the solvent effect of acetonitrile using Solvent Polarizable Continuum Model (PCM). Unless specified otherwise, the Gibbs free energy was used throughout. Considering the deviation in the free energies is ~1.89 kcal/mol from the standard state (1 atm) to 1 M in solution, we reduced by 1.89 kcal/mol to the free energy for additional steps and added by 1.89 kcal/mol for the dissociation steps[57]. For transition state, intrinsic reaction coordinate (IRC) calculations were performed to verify whether it

connected with correct reactants and products or intermediates. Time-dependent density functional theory (TD-DFT) was performed to calculate the vertical excitation energies of the photodissociation process, using the CAM-B3LYP-D3(BJ)[58] level of theory with the same basis set. All calculations were performed using the Gaussian 16 Rev. A.03 software suite[59]. The geometries were realized using CYLview, 1.0[60].

## Data availability

The authors declare that all other data supporting the findings of this study are available within the article and Supplementary Information files, and also are available from the corresponding author upon reasonable request.

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

## Acknowledgements

J.W. is grateful for the financial support provided by the Ministry of Education (MOE) of Singapore (MOE2017-T2-2-081), National University of Singapore (R-143-000-B60-114), NUS (Suzhou) Research Institute and National Natural Science Foundation of China (Grant No. 21702142, 21871205). We also express our gratitude to Prof. Zhi-Jian Zhao (Tianjin University) for helpful discussions regarding the DFT calculations and Dr. Tengfei Ji (RWTH Aachen University) for assistance on mechanism study.

## Author contributions

H.W. discovered and developed the reaction. H.W. and J.W. conceived and designed the investigations. X.S. conducted density functional theory (DFT) calculations. H.W., R.W.T., T.W., and X.C. performed the experiments. H.W., R.W.T. and J.W. wrote the paper.

## Competing interests

The authors declare no competing interests.
