## [Peer Review File · Nature Communications]

REVIEWER COMMENTS

Reviewer #1 (Remarks to the Author):

Wu and co-workers described a photo-induced deconstructive fragmentation of simple alkenes with water and halogen reagents in the absence of any external catalyst. β -Scission of the generated alcohol intermediates leads the generation of a wide range of geminal diiodoalkanes and bromo(iodo)alkanes, which are useful synthons in organic synthesis and are otherwise difficult to synthesize. One drawback of this work is the difficult transformation of trialkyl substituted alkenes (21% under 80 °C). However, the proposed generation of alkyl hypoiodites is unique and is supported by their control experiments and calculations. Taken together, this method is novel and the generated products are useful. I therefore recommend its publication in Nat. Commun. after addressing the following issues.

1. Cyclopentene and cycloheptene derivatives have been transferred to corresponding geminal diiodoalkanes. However, only cyclopentenes were presented in oxidative deconstructive geminal bromo-iodination. What about the results of cycloheptene derivatives in bromo-iodination?
2. Geminal dihalides are interesting and important compounds. Based on the reaction mechanism, is formation of geminal bromo-iodination, geminal chloro-iodination and fluoro-iodination possible? The authors should elaborate these potentials.
3. Trialkyl substituted alkenes did not work well. The author should provide more explanation on this observation.
4. Why there is no product with 44, but good yield with 45?
5. Ref. 21 should belong to oxidative cleavage, which is mis-assigned in the main text.

Reviewer #2 (Remarks to the Author):

This manuscript describes a method for photo-induced deconstructive geminal dihalogenation of alkenes. The cleavage of C=C bond of similar substrates via radical intermediate is known in the literature (ref. 18 and 21 of the manuscript). The reaction presented in this manuscript proceeds via in situ generated cycloalkanols where the cleavage of C-C single bond takes place with participation of reactive oxygen radical species. Similar cleavage process for C-C bond of cycloalkanols via a photo-induced radical pathway has already been reported by others (Angew. Chem. Int. Ed. 2019, 58, 2 – 13; Angew. Chem. Int. Ed. 2016, 55, 15319 –15322; ref. 26 of the manuscript). Recently, Wang reported a method for deconstructive halogenation of cycloalkanols (Chem. Commun, 2020, 56, 5002-5005). The use of dibromohydantoin for deconstructive halogenation of cyclic amines is known in the literature (Nature, 2018, 564, 244).

Although, this method reports a new synthetic route for geminal dihaloalkanes, the concept and strategy reported in this manuscript are not appreciably novel.

Moreover, this method has following limitations:

The reaction takes long irradiation hours (36h) under a very high power LED light (80 W) for achieving moderate yield of the products. In my opinion, temperature of the reaction mixture should go up significantly after irradiating for such as long time using a 80 W LED light.

Yields of halogenated products are not impressive. There should have been more experiments for improvement of the yield.

This reaction is limited to aryl substituted cyclic olefins only.

Hence, this manuscript is not suitable for publication in Nature Communications.

Reviewer #3 (Remarks to the Author):

Wu and co-workers developed an efficient photo-mediated deconstructive functionalization of carbon-carbon double bonds. Meanwhile, a protocol for construct geminal diiodination and bromo-iodination of trisubstituted alkenes under visible light irradiation has been developed, which represent a unique class of compounds. Efficient synthetic pathways for geminal dihalides are quite limited. Therefore, this is an interesting work. Publication in Nature Communication is recommended after minor revision:

"In Fig. 4a, trialkyl-substituted alkene 41 was subjected to the standard conditions for geminal diiodination and bromo-iodination, but no corresponding product could be generated. When the reaction temperature of bromo-iodination was raised to 80 °C, product 42 was isolated in 21% yield." The author mentioned "This result demonstrated the importance of aromatic substituents for the success of the deconstructive functionalization." But it looks that temperature is more important than aromatic substituents for this reaction. The authors should explain the phenomenon and their conclusion clearly.

According to Fig. 4a, when the reaction temperature was raised to 80 °C, could the reaction work by using acyclic trisubstituted olefins as substrate, such as 43 and 44? When the reaction temperature was raised to 80 °C, could the reaction work by using cyclohexene derivatives without aromatic ring substitution as the substrate?

Besides the generate of geminal diiodo and bromo(iodo) products, can this strategy be further expanded to produce chloro(iodo) products by addition NCS or DCDMH to this reaction?

In addition, it would be better to cite some related literatures: Chinese Journal of Chemistry, 2018, 36, 712; Org. Lett. 2012, 14, 4158.

Response to Reviewer # 1

Comment 1: *Wu and co-workers described a photo-induced deconstructive fragmentation of simple alkenes with water and halogen reagents in the absence of any external catalyst. β -Scission of the generated alcohol intermediates leads the generation of a wide range of geminal diiodoalkanes and bromo(iodo)alkanes, which are useful synthons in organic synthesis and are otherwise difficult to synthesize. One drawback of this work is the difficult transformation of trialkyl substituted alkenes (21% under 80 °C). However, the proposed generation of alkyl hypoiodites is unique and is supported by their control experiments and calculations. Taken together, this method is novel and the generated products are useful. I therefore recommend its publication in Nat. Commun. after addressing the following issues.*

Response to Comment 1: We thank Reviewer #1 for all the supportive comments and insightful suggestions.

Comment 2: *Cyclopentene and cycloheptene derivatives have been transferred to corresponding geminal diiodoalkanes. However, only cyclopentenenes were presented in oxidative deconstructive geminal bromo-iodination. What about the results of cycloheptene derivatives in bromo-iodination?*

Response to Comment 2: We have tested cycloheptene derivatives in bromo-iodination. However, the corresponding bromo-iodoalkane products could not be formed. We have updated the manuscript context and Fig. 3 accordingly.

Comment 3: *Geminal dihalides are interesting and important compounds. Based on the reaction mechanism, is formation of geminal bromo-iodination, geminal chloro-iodination and fluoro-iodination possible? The authors should elaborate these potentials.*

Response to Comment 3: We thank Reviewer #1 for this comment. The transformation is efficient with diiodination and geminal bromo-iodination. We have attempted geminal chloro-iodination and fluoro-iodination using Cl^+ reagents and F^+ reagents, including NCS, DCDMH (1,3-dichloro-5,5-dimethylhydantoin), BI-Cl (chloro benziodoxole), NFSI, and Selectfluor. However, when we used these reagents instead of NBS under the standard conditions, no desired product was obtained.

We speculated that compared with halohydroxylation of alkenes using NBS and NIS/DIH, halohydroxylation with Cl⁺ reagents and F⁺ reagents resulted in low yields of chlorohydrins and fluoro-hydrins (*Tetrahedron Letters* 2009, 50, 5754–5756). The inefficient generation of halohydrin intermediates caused no product formation in our cascade transformation.

Comment 4: Trialkyl substituted alkenes did not work well. The author should provide more explanation on this observation.

Response to Comment 4: We thank Reviewer #1 for this comment. The generation and stability of the halohydrin and hypoiodite intermediates are the keys to the success of geminal diiodination and bromo-iodination. Control experiments were performed to evaluate the temperature effect on the halohydrin formation and deconstructive iodination with both aryl substituted and trialkyl substituted cyclic alkenes (Supplementary Fig. S2). Although a high temperature (80 °C) could promote the bromo-iodination of trialkyl-substituted alkene **42**, diiodination of **42** was failed under the same conditions. This is due to that the intermediate alkyl substituted iodohydrin is not thermal stable. Even though the alkyl substituted bromohydrin is relatively more stable, it also decomposed at high temperatures. The decomposition of halohydrins explained the failure of diiodination of **42** at 80 °C and low yield of bromo-iodination of **42** at 100 °C.

We have revised the maintext: “A high temperature (80 °C) could not promote geminal diiodination of **42** either, while bromo-iodination product **43** could be isolated in 21% yield. However, further increasing the temperature to 100 °C afforded only a trace amount of **43**. Control experiments (Supplementary Fig. S2) indicated the instability of halohydrins at high temperatures and the ineffectiveness of trialkyl substituted halohydrins in the deconstructive iodination step. These results highlighted the importance of a suitable temperature and aromatic substituents for the success of the deconstructive geminal dihalogenation.” Fig. 4 and the Supplementary Information have been updated accordingly as well.

Comment 5: Why there is no product with 44, but good yield with 45?

Response to Comment 5: (Please note that compounds **44** and **45** were renumbered to **45** and **46**.) No product could be detected from **45** with DIH, but the intermediate iodohydrin was formed. The problem should be associated with hypoiodite intermediate formation from the corresponding iodohydrin with DIH. We proposed that the intermediate iodohydrin generated from **45** was too steric hindered to react with DIH efficiently.

Comment 6: 5. Ref. 21 should belong to oxidative cleavage, which is mis-assigned in the main text.

Response to Comment 6: We thank Reviewer #1 for pointing this out. References have been reorganized accordingly.

Response to Reviewer # 2

Comment 1: *This manuscript describes a method for photo-induced deconstructive geminal dihalogenation of alkenes. The cleavage of C=C bond of similar substrates via radical intermediate is known in the literature (ref. 18 and 21 of the manuscript). The reaction presented in this manuscript proceeds via in situ generated cycloalkanols where the cleavage of C-C single bond takes place with participation of reactive oxygen radical species. Similar cleavage process for C-C bond of cycloalkanols via a photo-induced radical pathway has already been reported by others (Angew. Chem. Int. Ed. 2019, 58, 2 – 13; Angew. Chem. Int. Ed. 2016, 55, 15319 –15322; ref. 26 of the manuscript). Recently, Wang reported a method for deconstructive halogenation of cycloalkanols (Chem. Commun, 2020, 56, 5002-5005). The use of dibromohydantoin for deconstructive halogenation of cyclic amines is known in the literature (Nature, 2018, 564, 244). Although, this method reports a new synthetic route for geminal dihaloalkanes, the concept and strategy reported in this manuscript are not appreciably novel.*

Response to Comment 1: We thank Reviewer #2 for these comments. However, we respectfully disagree with the comments which underestimate the core values of our work.

“The cleavage of C=C bond of similar substrates via radical intermediate is known in the literature (ref. 18 and 21 of the manuscript).” Both ref. 18 and ref. 21 represent nice studies on oxidative cleavage of C=C bond using air. Both reaction mechanisms and products are totally different from our transformation.

“Similar cleavage process for C-C bond of cycloalkanols via a photo-induced radical pathway has already been reported by others (Angew. Chem. Int. Ed. 2019, 58, 2 – 13; Angew. Chem. Int. Ed. 2016, 55, 15319 –15322; ref. 26 of the manuscript). Recently, Wang reported a method for deconstructive halogenation of cycloalkanols (Chem. Commun, 2020, 56, 5002-5005). The use of dibromohydantoin for deconstructive halogenation of cyclic amines is known in the literature (Nature, 2018, 564, 244).” We are unable to find the article based on “*Angew. Chem. Int. Ed. 2019, 58, 2 – 13*”. “*Angew. Chem. Int. Ed. 2016, 55, 15319 –15322*” reported the formation of oxyl radicals from cycloalkanols catalyzed by CeCl₃ under visible light irradiation, which was followed by β -scission and amination using di-tert-butyl azodicarboxylate. “ref 26” is a review paper which summarized recently developed methods for alkoxy radical generation and the subsequent C-C bond scission. “*Chem. Commun, 2020, 56, 5002-5005*” describes a nice example of ring-opening iodination and bromination of cycloalkanols in the presence of an excess amount of PIDA (I⁺³ reagent) as the oxidant. “*Nature, 2018, 564, 244*” reported the deconstructive cleavage of cyclic amines through the formation of hemiaminal. All these reports represent nice studies of alkoxy radical formation from alcohols and subsequent bond scission. However, the conditions are normally limited to strong oxidants such as silver reagents and I⁺³ reagents (e.g. PIDA).

In our study, we disclosed a novel process to form alkyl hypoiodites from benzyl alcohols and stable I⁺¹ reagents (*N*-iodoimides), which has not been reported before. These mild reagents can also be applied to halohydration of alkenes to allow selective deconstructive functionalization of C=C double bonds in a cascade fashion. A wide range of geminal diiodoalkanes and bromo(iodo)alkanes (>40 examples) were directly prepared from various trisubstituted alkenes, which are otherwise difficult to access.

We have included these literatures in our updated references.

Comment 2: *Moreover, this method has following limitations: The reaction takes long irradiation hours (36h) under a very high power LED light (80 W) for achieving moderate yield of the products. In my opinion, temperature of the reaction mixture should go up significantly after irradiating for such as long time using a 80 W LED light.*

Response to Comment 2: We thank Reviewer #2 for those comments. Our transformations are cascade reactions, and we conducted 36 hours to ensure all the substrates in the scope evaluation could achieve full conversion. The light on/off experiment of deconstructive diiodination of compound **1** (Fig. 5c) also indicates that the 36 hours reaction time is essential to achieve high conversion.

We used two Kessil LED lamps (40 W x 2) for our transformations, and the distance between the LED lamps and the reactor is 8 cm. The temperature could be controlled at 30 °C when a cooling fan was used. Without the cooling fan, the temperature was maintained around 50 °C, which was applied as our standard reaction conditions. The higher temperature (80 °C, 100 °C) was controlled by a water or oil bath with a heating plat equipped with digital temperature control.

We have updated these details in Supplementary Information (Fig S1).

Comment 3: *Yields of halogenated products are not impressive. There should have been more experiments for improvement of the yield.*

Response to Comment 3: We have optimized our reaction conditions as illustrated in Table 1. Even though we did not optimize individual substrates, the majority of dehalogenation products were obtained in the range of 60% to 85% yield. The intermediate halohydrins are not very stable. Considering this is a cascade transformation, the yields are actually descent. These dihalide compounds are versatile intermediates which can be further diversified. We provided direct access to these useful compounds which are otherwise difficult to prepare.

Comment 4: *This reaction is limited to aryl substituted cyclic olefins only.*

Response to Comment 4: Control experiments were performed to evaluate the halohydrin formation and deconstructive iodination with trialkyl substituted cyclic alkenes (Supplementary Fig. S2). Although a high temperature (80 °C) could promote the bromo-iodination of trialkyl-substituted alkene **42**, diiodination of **42** was failed under the same conditions. This is due to the intermediate alkyl substituted iodohydrin is not thermal stable. Even though the alkyl substituted bromohydrin is relatively more stable, it also decomposed at high temperatures. The decomposition of halohydrins explained the failure of diiodination of **42** at 80 °C and low yield of bromo-iodination of **42** at 100 °C.

We have revised the maintext: “A high temperature (80 °C) could not promote geminal diiodination of **42** either, while bromo-iodination product **43** could be isolated in 21% yield. However, further increasing the temperature to 100 °C afforded only a trace amount of **43**. Control experiments (Supplementary Fig. S2) indicated the instability of halohydrins at high temperatures and the ineffectiveness of trialkyl substituted halohydrins in the deconstructive iodination step. These

results highlighted the importance of a suitable temperature and aromatic substituents for the success of the deconstructive geminal dihalogenation.” Fig. 4 and the Supplementary Information have been updated accordingly as well.

Response to Reviewer # 3

***Comment 1:** Wu and co-workers developed an efficient photo-mediated deconstructive functionalization of carbon-carbon double bonds. Meanwhile, a protocol for construct geminal diiodination and bromo-iodination of trisubstituted alkenes under visible light irradiation has been developed, which represent a unique class of compounds. Efficient synthetic pathways for geminal dihalides are quite limited. Therefore, this is an interesting work. Publication in Nature Communication is recommended after minor revision:*

Response to Comment 1: We thank Reviewer #3 for the supportive comments and valuable suggestions.

***Comment 2:** “In Fig. 4a, trialkyl-substituted alkene 41 was subjected to the standard conditions for geminal diiodination and bromo-iodination, but no corresponding product could be generated. When the reaction temperature of bromo-iodination was raised to 80 °C, product 42 was isolated in 21% yield.” The author mentioned “This result demonstrated the importance of aromatic substituents for the success of the deconstructive functionalization.” But it looks that temperature is more important than aromatic substituents for this reaction. The authors should explain the phenomenon and their conclusion clearly.*

Response to Comment 2: We thank Reviewer #3 for this suggestion. (Please note that compounds **41** and **42** were renumbered to **42** and **43**.) The generation and stability of the halohydrin and hypoiodite intermediates are the keys to the success of geminal diiodination and bromo-iodination. Control experiments were performed to evaluate the temperature effect on the halohydrin formation and deconstructive iodination with both aryl substituted and trialkyl substituted cyclic alkenes (Supplementary Fig. S2). Although a high temperature (80 °C) could promote the bromo-iodination of trialkyl-substituted alkene **42**, diiodination of **42** was failed under the same conditions. This is due to that the intermediate alkyl substituted iodohydrin is not thermal stable. Even though the alkyl substituted bromohydrin is relatively more stable, it also decomposed at high temperatures. The decomposition of halohydrins explained the failure of diiodination of **42** at 80 °C and low yield of bromo-iodination of **42** at 100 °C.

We have revised the maintext: “A high temperature (80 °C) could not promote geminal diiodination of **42** either, while bromo-iodination product **43** could be isolated in 21% yield. However, further increasing the temperature to 100 °C afforded only a trace amount of **43**. Control experiments (Supplementary Fig. S2) indicated the instability of halohydrins at high temperatures and the ineffectiveness of trialkyl substituted halohydrins in the deconstructive iodination step. These results highlighted the importance of a suitable temperature and aromatic substituents for the success of the deconstructive geminal dihalogenation.” Fig. 4 and the Supplementary Information have been updated accordingly as well.

Comment 3: According to Fig. 4a, when the reaction temperature was raised to 80 °C could the reaction work by using acyclic trisubstituted olefins as substrate, such as 43 and 44? When the reaction temperature was raised to 80 °C could the reaction work by using cyclohexene derivatives without aromatic ring substitution as the substrate?

Response to Comment 3: Thank Reviewer #3 for these suggestions. (Please note that compounds 43 and 44 were renumbered to 44 and 45.) Product 47 could not be obtained from 44 and 45 at 80 °C. As we demonstrated in Figure S2 (Supplementary Information), the iodohydrins are very unstable under heating. We tried diiodination and bromo-iodination of compound 42, a trialkyl substituted cyclohexene derivative, at various reaction temperatures. Due to the instability of iodohydrins and bromohydrins, the only successful example is the formation of product 43 as illustrated in Fig. 4a.

Comment 4: Besides the generate of geminal diiodo and bromo(iodo) products, can this strategy be further expanded to produce chloro(iodo) products by addition NCS or DCDMH to this reaction?

Response to Comment 4: We thank Reviewer #3 for this suggestion. We have attempted geminal chloro-iodination and fluoro-iodination using Cl⁺ reagents and F⁺ reagents, including NCS, DCDMH (1,3-dichloro-5,5-dimethylhydantoin), BI-Cl (chloro benziodoxole), NFSI, and Selectfluor. However, when we used these reagents instead of NBS under the standard conditions, no desired product was obtained. We speculated that compared with halohydroxylation of alkenes using NBS and NIS/DIH, halohydroxylation with Cl⁺ reagents and F⁺ reagents resulted in low yields of chlorohydrins and fluoro-hydrins (*Tetrahedron Letters* 2009, 50, 5754–5756). This inefficient generation of halohydrin intermediates caused no product formation in our cascade transformation.

Comment 5: In addition, it would be better to cite some related literatures: Chinese Journal of Chemistry, 2018, 36, 712; Org. Lett. 2012, 14, 4158.

Response to Comment 5: We have included these related literatures in the updated references.

REVIEWERS' COMMENTS:

Reviewer #1 (Remarks to the Author):

I have checked the revision carefully; all my scientific concerns have been well addressed. I do not have any more questions. Thus, I support its publication in Nat. Commun. as is.

Response to Reviewer #1:

Reviewer #1 (Remarks to the Author):

I have checked the revision carefully; all my scientific concerns have been well addressed. I do not have any more questions. Thus, I support its publication in Nat. Commun. as is.

Response: Thanks for the reviewer's supporting comments.